# The Oocyte-Specific Linker Histone H1FOO Is Not Essential for Mouse Oogenesis and Fertility

**DOI:** 10.3390/cells11223706

**Published:** 2022-11-21

**Authors:** Fernando Sánchez-Sáez, Raquel Sainz-Urruela, Natalia Felipe-Medina, Yazmine B. Condezo, Manuel Sánchez-Martín, Elena Llano, Alberto M. Pendás

**Affiliations:** 1Molecular Mechanisms Program, Centro de Investigación del Cáncer, Instituto de Biología Molecular y Celular del Cáncer, CSIC-Universidad de Salamanca, 37007 Salamanca, Spain; 2Departamento de Medicina, Universidad de Salamanca, 37007 Salamanca, Spain; 3Departamento de Fisiología y Farmacología, Universidad de Salamanca, 37007 Salamanca, Spain

**Keywords:** fertility, oogenesis, meiosis, linker histone H1FOO

## Abstract

Meiosis is a highly conserved specialized cell division process that generates haploid gametes. Many of its events are associated with dynamically regulated chromosomal structures and chromatin remodeling, which are mainly modulated by histone modifications. Histone H1 is a linker histone essential for packing the nucleosome into higher-order structures, and H1FOO (H1 histone family, member O, oocyte-specific) is a H1 variant whose expression pattern is restricted to growing oocytes and zygotes. To further explore the function of H1FOO, we generated mice lacking the *H1foo* gene by the CRISPR/Cas9 technique. Herein, we combine mouse genetics and cellular studies to show that *H1foo*-null mutants have no overt phenotype, with both males and females being fertile and presenting no gross defects in meiosis progression nor in synapsis dynamics. Accordingly, the histological sections show a normal development of gametes in both male and female mice. Considering the important role of oocyte constituents in enhancing mammalian somatic cell reprogramming, we analyzed iPSCs generation in *H1foo* mutant MEFs and observed no differences in the absence of H1FOO. Taken all together, in this work we present the first in vivo evidence of H1FOO dispensability for mouse fertility, clarifying the debate in the field surrounding its essentiality in meiosis.

## 1. Introduction

Gametogenesis is defined by a unique and highly dynamic program of events that results in the generation of haploid gametes [1]. This reduction in the genetic content requires the precise regulation of several processes including homologous chromosome synapsis, crossover (CO) formation, subsequent homolog segregation, and chromatin remodeling [2].

Oocyte meiotic maturation is a complex and vital process necessary to attain full oocyte competence, as well as for a proper early embryonic development [3]. In contrast to most of the other meiotic processes, female mammalian meiosis undergoes two different arrests during oocyte maturation. First, the developing oocytes arrest before birth at the diplotene stage with a germinal vesicle (GV) in their nucleus. This state is maintained until meiosis resumption at puberty in response to luteinizing hormones (LHs). Meiosis resumes after GV breakdown (GVBD), completing prophase I and progressing until the second arrest at metaphase II. Finally, female meiosis is completed after fertilization [4].

In the nucleus, DNA is packaged in fundamental subunits named nucleosomes. Histones are basic nuclear proteins that play a fundamental role in the generation of the nucleosome structure of the chromosomal fiber in eukaryotes. Nucleosomes consist of approximately 146 bp of DNA wrapped around a histone octamer composed of pairs of each of the four core histones (H2A, H2B, H3, and H4) [5]. In eukaryotes, a fifth class of histones termed H1 acts as a linker histone, binding to the DNA between nucleosomes and promoting a higher order of chromatin organization [6]. This is the most heterogeneous family of histones, consisting of eleven different variants in mammals: seven somatic H1s (H1^0^, H1a, H1b, H1c, H1d, H1e, and H1x) and four germ-specific H1s (the testis-specific H1t, H1T2, and H1LS1 and the oocyte-specific H1FOO) [7,8,9]. Largely, H1 molecules consist of a highly conserved central globular domain with more variable tail regions at both their N- and C-terminal ends [10]. In addition to their structural role, H1 histones may be involved in gene expression regulation. Nonetheless, the aforementioned existence of multiple H1 subtypes and post-translational modifications hampers the study of the roles that this protein family might play in heterochromatin formation, transcriptional regulation, and embryogenesis.

H1FOO (H1 histone family, member O, oocyte-specific) is a replication-independent histone that is a member of the histone H1 family, a mammalian homolog of the oocyte-specific linker histone B4 of the frog, and of the cs-H1 linker histone of the sea urchin [11]. Unlike most intronless histone’s genes, the single-copy *H1foo* gene contains five exons and is highly expressed in oocytes and early embryos [12]. H1FOO presents a N-terminal domain containing multiple potential phosphorylation sites, and a long C-terminal tail rich in acidic amino acids [11]. Both its N-terminal and globular domains are required for the correct association with chromatin in the oocyte nucleus [13]. Functionally, it has been suggested to play a role in the regulation of the chromosomal fiber. *H1foo* knockdown in one-cell-stage embryos leads to a tighter state of the chromatin structure in the pronucleus and to an increase in the deposition of the histone H3 variant H3.1/3.2 in the peripheral region of the pronucleus [14]. In addition to this, the implication of H1FOO in oocyte maturation is still under strong debate [15,16,17]. Due to this, the in vivo analysis of H1FOO deficiency could shine light on the molecular mechanisms that regulate the early steps in gametogenesis [18].

After fertilization, and in order to generate a totipotent embryonic genome, maternal and paternal chromatins are required to be comparably ordered. To do so, paternal chromatin undergo an extensive decondensation whilst protamines are replaced by maternal core histones and H1FOO [13,19], facilitating nuclear reprogramming. Eventually, the two pronuclei fuse, and by the end of the two-cell stage, H1FOO is replaced by somatic H1s to meet the requirements of the embryonic transcriptional program.

Histones are also involved in epigenetic modifications that occur during induced cellular reprogramming, since it may either promote or inhibit gene transcription by altering chromatin folding [20]. As a response to ectopic expression of the reprogramming factors, chromatin remodeling takes place [21] and, although the dynamics of somatic H1s differentiation and reprogramming are described [22], H1FOO involvement in vivo in these processes is still to be assessed in detail.

Given the absence of functional studies, we address the first in vivo functional analysis of H1FOO-deficient mice. Homozygous mutant mice show no overt phenotype, with both males and females being completely fertile. Accordingly, the histological sections reveal a normal development of gametes in female mutant mice and the fertility assessment is similar to the wild-type one. Analyzed in greater detail, oogenesis shows no defects in synapsis nor in reaching the first meiotic arrest. Moreover, the absence of H1FOO in mouse embryonic fibroblasts (MEFs) does not reduce the reprogramming efficiency of somatic cells to induced pluripotent stem cells (iPSCs). Taken all together, our results represent the first in vivo evidence of the lack of essentiality of H1FOO for mouse gametogenesis.

## 2. Materials and Methods

### 2.1. Mice

In order to develop the mutant mouse model (*H1foo^−/−^*), making use of CRISPR/Cas9 genome editing technique, crRNAs were predicted at the website https://eu.idtdna.com/site/order/designtool/index/CRISPR_CUSTOM (accessed on 20 September 2017). The crRNAs, the tracrRNA, and the single single-stranded oligodeoxynucleotides (ssODNs) were produced by chemical synthesis at IDT (crRNAs and ssODN sequences are listed in Appendix A). The crRNAs and tracrRNA were annealed to obtain the mature sgRNA. A mixture containing the sgRNAs, recombinant Cas9 protein (IDT), and the ssODN was microinjected into F2 zygotes (hybrids between strains C57BL/6J and CBA/J) at the Transgenic Facility of the University of Salamanca. Edited founders were identified by polymerase chain reaction (PCR) amplification (Taq polymerase, NZYTech) with primers listed in Appendix A, producing amplicons of 272 or 268 base pairs (bp) for either wild-type (WT) or edited alleles, respectively. These amplicons were subcloned into pBlueScript (Stratagene), followed by Sanger sequencing for a verification in detail of the editing. Selected founders, carrying the desired alleles, were crossed with wild-type mice to eliminate possible unwanted off-targets. Heterozygous mice were re-sequenced and crossed to generate the edited homozygotes.

### 2.2. Fertility Assessment

For the experiment, 8 week old *H1foo^+/−^* males were mated with *H1foo^+/−^* females over the course of 4–12 months, and similarly, distinct matings of *H1foo^−/−^* males with *H1foo^−/−^* females were set. In order to further evaluate the existence of a delay phenotype in successive generations, a *H1foo^−/−^* F2 male was mated with two *H1foo^−/−^* F2 females for five months. The presence of copulatory plug was examined daily, and the number of pups per litter was recorded.

### 2.3. Histology

For the histological analysis, adult ovaries were fixed in 10% formol during 24 h at room temperature. After that, they were processed into serial paraffin sections, and stained with hematoxylin–eosin. The samples were analyzed using a microscope OLYMPUS BX51 and images were taken with a digital camera OLYMPUS DP70.

### 2.4. Immunocytology and Antibodies

Testes were detunicated and processed for spreading using a conventional “dry-down” technique [23]. Oocytes from 16.5 days post coitum (dpc) fetal ovaries were digested with collagenase, incubated in hypotonic buffer, disaggregated, and fixed in 40 µL of fixative buffer (1% (*w*/*v*) paraformaldehyde, 5 mM sodium borate, 0.15% Triton X-100, 3 mM DTT, pH 9.2). The meiocyte preparations were incubated with the following primary antibodies, diluted in 1x phosphate-buffered saline (PBS), for immunofluorescence (IF): rabbit αH1FOO serum (kindly provided by Dr. J. D. Hennebold, 1:20), mouse αSYCP3 IgG sc-74569 (Santa Cruz Biotechnology, Dallas, TX, USA, 1:1000), rabbit αSYCP1 IgG ab15090 (Abcam, Cambridge, United Kingdom, 1:200), rabbit anti-γH2AX (ser139) IgG #07–164 (Millipore, Burlington, MA, USA, 1:200), ACA or purified human α-centromere proteins IgG 15–235 (Antibodies Incorporated, Davis, CA, USA, 1:5), rabbit αSUN1 M-300 sc-135075 (Santa Cruz, 1:25). The secondary antibodies used were goat Alexa 555 α-mouse A-32727, goat Alexa 488 α-mouse A-11001, donkey Alexa 555 α-rabbit A-31572, (Thermo Fisher, Waltham, MA, USA, 1:200), goat Alexa 488-Fab α-rabbit 111-547-003, and donkey TRITC α-human 709-025-149 (Jackson Immunoresearch, Cambridge, United Kingdom, 1:100). Slides were mounted with Vectashield mounting medium supplemented with DAPI and visualized at room temperature using a Leica DM6000b microscope with an 63x objective. Images were taken with a digital camera (ORCA-ER C4742-80; Hamamatsu) and processed with Leica LAS X Life Science Software and Adobe Photoshop CS6 (Adobe, San José, CA, USA). Quantification of fluorescent signaling was performed using ImageJ 1.52a software (Wayne Rasband, National Institutes of Health, USA).

### 2.5. Unfertilized Oocyte Collection for Metaphase II Analysis

Ovaries from 53 day old unstimulated females were removed and oocytes were subsequently released by puncturing ovaries using 30-gauge needles. Primary oocytes were cultured in KSOM medium (Millipore, Burlington, MA, USA) and covered with a drop of mineral oil for 18 h at 37 °C in order to let them reach the metaphase II stage. Afterwards, cell suspension was fixated in a slide with the fixative buffer previously described in Materials and Methods (Section 2.4.), and the corresponding αH1FOO IF was performed.

### 2.6. Cell Culture

Primary MEFs were derived from embryonic day 13.5 (E13.5) embryos following standard procedures. MEFs and HEK293T (obtained from the American Type Culture Collection, ATCC) cell lines were cultured at atmospheric oxygen pressure in Dulbecco’s modified Eagle’s medium (GIBCO, Thermo Fisher) supplemented with 10% fetal bovine serum (GIBCO, Thermo Fisher), and 2 mM glutamine. HEK293T cells were transfected with Jetpei (PolyPlus) according to the manufacturer’s protocol. Cell lines were tested for mycoplasma contamination using the Mycoplasma PCR ELISA (Sigma-Aldrich, St. Louis, MO, USA).

### 2.7. iPSCs Generation from MEFs

For iPSCs generation, 2.5 × 10^5^ MEFs were infected with retroviral particles produced by HEK293T transfected with constitutive retroviral expression vectors pMXs KLF4, OCT4, and SOX2. The iPSC media (DMEM, GIBCO; 15% KSR, Invitrogen, Thermo Fisher; 1% non-essential amino acids, MEM NEE 100X GIBCO; 1% PSG; 0.002% β-mercaptoethanol 50 mM, GIBCO; 1000 units/mL LIF, Merck, Darmstadt, Germany) was changed every 24 h until iPSC colonies appeared (after ∼14 days of treatment). Three weeks after plating the MEFs, reprogramming plates were stained for alkaline phosphatase activity (AP detection kit, Merck).

### 2.8. Statistics

In order to compare counts between genotypes, we used the Welch´s *t*-test (unequal variances *t*-test), which was appropriate as the count data were not highly skewed (i.e., were reasonably approximated by a normal distribution) and, in most cases, showed unequal variance. We applied a two-sided test in all the cases. N.s. (not significant): *p*-value ≥ 0.05.

### 2.9. Ethics Statement

All the experiments were approved by the Ethics Committee for Animal Experimentation of the University of Salamanca (USAL) and the Ethics Committee of the Spanish Research Council (CSIC) under protocol #00–245. Accordingly, all the mouse protocols used in this work were approved by the Animal Experimentation committees mentioned above. Specifically, mice were always housed in a temperature-controlled facility (specific pathogen free, SPF) using individually ventilated cages, standard diet, and a 12 h light/dark cycle, according to EU law (63/2010/UE) and the Spanish royal law (53/2013) at the “Servicio de Experimentación Animal”, SEA. In addition, animal suffering was always minimized, and we made every effort to improve animal welfare during the life of the animals.

## 3. Results

### 3.1. Generation of H1FOO Knockout Mice

Murine *H1foo* gene is encoded by five exons at chromosome 6, with the ATG codon located in the first exon and the STOP codon in the fifth. To gain further insight into H1FOO function, we generated a mouse model by CRISPR–Cas9 genome editing harboring a deletion of exons 2 and 3, which expectedly gives rise to a null allele (deletion of 70% of the coding sequence) (Figure 1a). A selected founder carrying the desired edition was crossed with WT C57BL/6J mice, and the heterozygous offspring were intercrossed to obtain the homozygous mutant mice, which were identified by PCR (Figure 1a).

To confirm whether the generated mutation was in fact a null allele, we analyzed the expression of H1FOO in unfertilized oocytes using an anti-H1FOO antibody. We observe H1FOO signal decorating the chromatin in WT unfertilized metaphase II (MII) oocytes (Figure 1b). The total absence of this signal in the knockout (KO) MII oocytes (Figure 1b) led us to validate our murine model. Taking into account the fact that basal *H1foo* transcription is observed in primary spermatocytes and oocytes [24], we first tried to immunolocalize H1FOO during prophase I using 16.5 dpc oocytes. Remarkably, a total absence of H1FOO labeling is detected in prophase I oocytes (Figure 1c). Considering H1FOO expression in human and mouse testes (www.proteinatlas.org accessed on 20 September 2017, [24]), and its load to sperm chromatin after fertilization, we sought to further explore its function in male gametogenesis, for which we carried out an immunofluorescence in surface-spread nuclei from mouse spermatocytes. However, we do not detect H1FOO labeling at any meiotic stage in spermatocytes (Appendix A).

### 3.2. H1FOO-Deficient Mice Are Fertile

Both male and female lacking H1FOO develop normally and show no overt phenotype. As H1FOO is known to be involved in meiotic maturation of mouse oocytes, we perform an evaluation of histological sections of both WT and KO ovaries. Adult *H1foo*-deficient ovaries appear to be normal, with no remarkable alteration in the distribution of follicles nor aberrant appearance of the stroma (Figure 2a). Additionally, adult male KO testis size and weight is similar to their WT counterparts (Figure 2b). To determine the potential effect of H1FOO absence on mice fertility, we performed a fertility assay. The presence of the copulatory plug was monitored daily to discard behavioral defects, and the number of pups per litter was recorded. Homozygous *H1foo* KO mice breeding shows similar litters in size and number to heterozygous mice mating (Figure 2c), which led us to discard an unusual mortality rate or any developmental defect in mutant embryos. Moreover, results from F2 KO breeding show an absence of a delayed phenotype in the next generations (Appendix A).

### 3.3. H1FOO-Deficient Meiocytes Show Normal Synapsis, Double-Strand Breaks (DSBs) Generation, and Telomere Dynamics

Given the transcription of *H1foo* in mouse spermatogonia and pachynema [24], and despite our negative immunolabeling of H1FOO in mouse spermatocytes and spermatogonia from chromosome spreads preparation, we analyzed meiotic progression in order to discard technical reasons (e.g., fixation) responsible for our inability to detect H1FOO in prophase I. To do so, we monitored the distribution of the central element protein SYCP1 and the lateral element SYCP3 labeling. We did not detect significative alteration in the synapsis/desynapsis dynamics of the synaptonemal complex (SC) in spermatocytes (Appendix A).

To further characterize the involvement of H1FOO in female meiotic progression, we analyzed the dynamics of assembly and disassembly of the SC in 16.5 dpc oocytes. An effective synapsis and desynapsis of the SC are observed in mutant oocytes (Appendix A), although a faint, yet not significant, delay at the zygotene stage is detected in *H1foo*-deficient oocytes (Appendix A, lower plot).

One of the major events occurring in early prophase I is the formation and repair of double-strand breaks (DSBs). Their generation relies on an ATM-dependent phosphorylation of histone H2AX at serine 139 (γ-H2AX), which activates the DSBs repair response at early meiotic prophase I. To thoroughly analyze a potential deregulation of DSBs generation that could be responsible for the subtle zygotene delay observed in the mutant oocytes, we immunostained γ-H2AX in 16.5 dpc oocytes. No significant differences are observed between genotypes (Appendix A).

In the search of any other mechanism underlying the narrow phenotype observed at the zygotene stage, and in order to discard any non-homologous telomeres fusion, we labelled oocytes telomeres by SUN1 and centromeres by anti-centromere antibodies (ACA). We do not detect any disturbance in telomere or centromere dynamics in *H1foo*-deficient oocytes at the zygotene or the pachytene stage (Appendix A). Taken all together, these observations suggest that H1FOO has no critical role in the progression of either male or female meiosis.

### 3.4. The Absence of H1FOO Does Not Impair MEFs Reprogramming

Somatic cells can be reprogrammed to iPSCs by ectopic expression of OCT4, SOX2, KLF4, and MYC (OSKM) [25], launching a cascade of events that includes a rearrangement of the epigenetic profile [26]. Oocyte constituents are involved in enhancing mammalian somatic cell reprogramming by somatic cell nuclear transfer (SCNT), as several candidate oocyte reprogramming factors (CORFs), including H1FOO, have been previously identified based on global transcriptional analysis [27]. Considering the increased efficiency of reprogramming after H1FOO overexpression [28], we evaluated the consequences of the loss of H1FOO expression in cell reprogramming using *H1foo^+/+^*, *H1foo^+/−^*, and *H1foo^−/−^* MEFs. The number of alkaline phosphatase (AP) positive colonies, typically used as a marker of undifferentiated cells, is similar in the three genotypes (Figure 3). In the light of this, and although increased *H1foo* expression contributes to the generation of higher quality iPSCs [28], its absence does not represent a significant hindrance to MEFs reprogramming to iPSCs.

## 4. Discussion

In the present study, we determined, for the first time, the lack of essentiality of the linker histone H1FOO for mouse fertility and development, using null mutant mice as a model. By carrying out a comprehensive biological analysis, here we show that the loss of this linker histone can be tolerated in vivo during oogenesis, as no significant obstacle for its progression occurs in homozygous mutant mice.

In order to analyze the expression pattern of H1FOO in mouse oocytes, we performed several immunolabeling experiments in oocytes at different stages. By doing so, we described H1FOO decorating the chromatin in WT metaphase II oocytes. The absence of signal in *H1foo*-deficient chromosomes led us to validate our murine model. Despite the fact that *H1foo* is transcribed during prophase I, our inability to stain H1FOO at any point during these stages and the absence of any subtle meiotic phenotype (see below) point towards an uncoupling of transcription and translation during the second meiotic division in females and during later stages in males. This translational repression is a cellular mechanism largely studied in mouse, essential for proper gametogenesis and embryo development [29,30]. This is in concordance with the very first description of mammalian H1FOO [11], with this histone being detected from the germinal vesicle stage to two-cell-stage embryos as a consequence of a highly regulated expression program that includes DNA methylation upstream of the *H1foo* gene [31].

Growing oocytes present a characteristic and strictly regulated loosened chromatin structure, which is needed for the acquisition of totipotency after fertilization. *H1foo*, among other factors such as the helicase *Chd9*, is believed to play a fundamental role in maintaining this structural regulation of the chromatin [28,32]. The change from loose to tight chromatin structure occurring in the one-cell to two-cell stage transition is responsible for the loss of totipotency of the embryo [33]. For this reason, it has been hypothesized that *H1foo* reaches a high expression level at the one-cell stage and it is dramatically reduced in two-cell stage embryos [14]. At this point, *H1foo* knockdown decreases the looseness of the chromatin structure and delays the completion of DNA synthesis in one-cell stage embryos [14]. Nevertheless, and according to our results, none of these alterations seem to be a major disturbance for the progression of murine preimplantation embryos in vivo, as no differences in either the histological analysis or in the number and size of litters are observed in homozygous mutant mice. This is in contrast with previous in vitro studies contemplating the essentiality of H1FOO for meiotic maturation of mouse oocytes, as its inhibition via antisense morpholino oligonucleotides (microinjected into germinal vesicle stage oocytes) led to failed first polar body extrusion and metaphase I arrest [15]. In a similar way, bovine oocyte maturation was also demonstrated to be hampered in vitro after siRNA-mediated *H1foo* inhibition [17]. In contrast, a recent study showed that *H1foo* knockdown using siRNAs had no effect on oocyte growth, maturation, and fertilization [18]. Taken together, we can conclude that, in opposition to some previous in vitro assays and in concordance with the most recent research lines, the absence of H1FOO can be tolerated in vivo with the specific underlying compensatory mechanisms still to be further elucidated.

Meiotic prophase I is the longest and most complex stage of the gametogenic program, as it encompasses a thoroughly regulated sequence of events, including homologous pairing, and DSB generation and their processing to COs. According to its later expression pattern, *H1foo* deficiency should not be a hindrance to early meiosis progression, as our observations reveal normal synapsis and telomer dynamics along prophase I. Bearing in mind that human *H1FOO* is transcribed in testes during spermatogenesis, and that H1FOO is quickly assembled into the introduced paternal genome after fertilization replacing the sperm-specific histone-like proteins [34,35], we sought to evaluate spermatogenesis progression under *H1foo* absence in order to discard a potential sexual dimorphism in this mutant. As expected, a male lacking H1FOO successfully reached the spermatozoa stage without any disturbance, being completely fertile.

The oocyte has the capacity to give rise to undifferentiated embryonic cells although it is a differentiated cell. This reprogramming implies a switch from somatic to oocyte transcriptional components. *Xenopus* histone B4 (the homolog of murine H1FOO) was demonstrated to be required for successful transcriptional reprogramming, probably by making somatic chromatin accessible to the oocyte machinery and subsequently enabling high amounts of Pol II loading [36]. It has been previously hypothesized that the absence of *H1foo* might reduce the efficiency of chromatin reprogramming in zygotes [7], even though the surviving *H1foo^−/−^* embryos and mice are completely viable. This fact could be explained by the activation of a compensatory mechanism that increases protein synthesis of other H1 variants to face the lack of H1FOO. Oocyte constituents play a crucial role in reprogramming in somatic cell nuclear transfer (SCNT), a procedure similar to iPSCs generation. Moreover, a rapid exchange of somatic linker histone with H1FOO in the chromatin of an injected somatic nucleus occurs during SCNT, a process analogous to the histone replacement occurring after fertilization. In this regard, *H1foo* exogenous expression was proven to enhance the number and quality of mouse iPSCs when co-expressed with *Oct4*, *Sox2,* and *Klf4* [28]. Both in vitro and in vivo differentiation potential of the iPSCs is enhanced after *H1foo* overexpression. In addition, ectopic expression of *H1foo* has been demonstrated to prevent normal differentiation into embryoid bodies, illustrating the impact of this histone on the epigenetic status [37]. This is in contrast to our observations, as MEFs lacking *H1foo* do not significantly affect iPSCs generation, with the number of AP colonies being even higher in the mutant than in the WT condition. This surprising result could be explained by a countervailing mechanism driven by other constituents that cope with the loss of this linker histone during reprogramming, as not only H1FOO but also somatic histones are able to associate with chromatin in the nuclei of somatic cells [13,38]. Pooling all these observations, we can conclude that H1FOO is a dispensable factor for iPSCs reprogramming.

Taken all together, in this study we gained further insights in H1FOO function and dispensability for mouse fertility and iPSCs reprogramming, contributing to elucidating processes and regulatory mechanisms of chromatin remodeling during mammalian oogenesis and enlightening the still open debate about H1FOO essentiality in meiosis.

## Figures and Tables

**Figure 1 cells-11-03706-f001:**
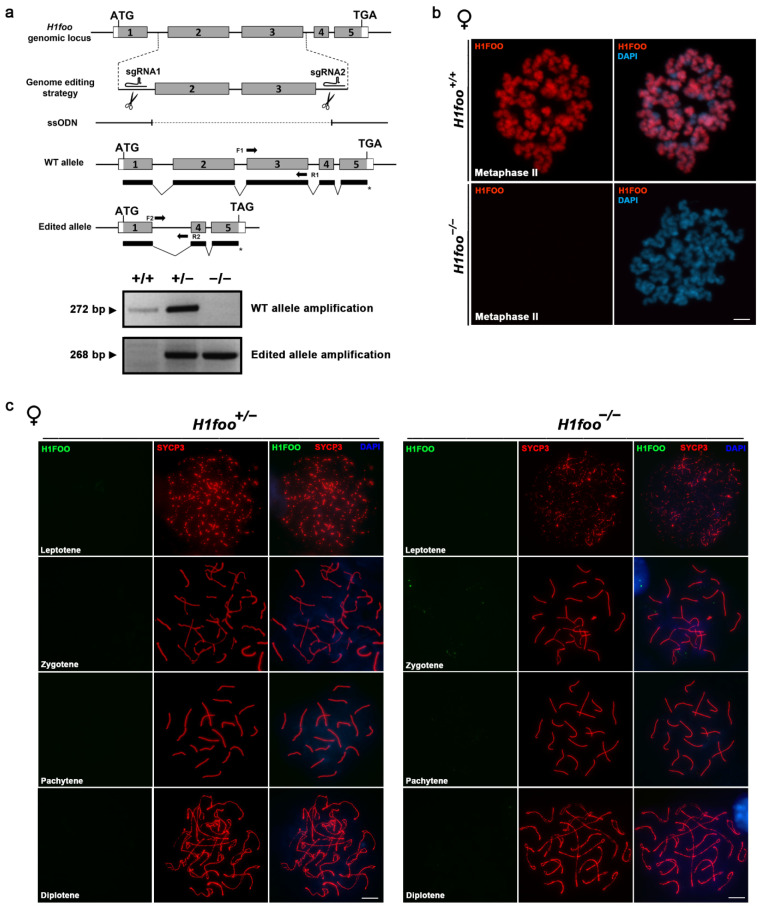
(**a**) Schematic representation of *H1foo* KO generation by CRISPR–Cas9 genome editing. Two sgRNAs flanking coding exons 2 and 3, respectively, of the *H1foo* genomic locus are represented. Coding (grey) and non-coding (open box) regions are indicated. Thick lines under the exons represent the expected transcripts derived from WT and *H1foo* edited alleles, with the * symbol pointing the translation termination. ATG, start codon; TGA, stop codon. Oligonucleotide sequences employed for the amplification of both WT (F1 and R1) and edited alleles (F2 and R2) are also pointed out. PCR analysis of genomic DNA from three *H1foo^+/+^*, *H1foo^+/−^*^,^ and *H1foo^−/−^* pups is shown (lower panel). (**b**) H1FOO immunolabeling in cultured unfertilized metaphase II oocytes from both *H1foo^+/+^* and *H1foo^−/−^* females showing H1FOO signal (red) decorating the chromosomes of WT, but not mutant oocytes. (**c**) Double immunolabeling of H1FOO (green) and SYCP3 (red) in fetal 16.5 dpc oocytes showing a complete absence of H1FOO signal along meiotic prophase I in both heterozygous and KO oocytes. Bars in panels (**b**,**c**), 10 µm.

**Figure 2 cells-11-03706-f002:**
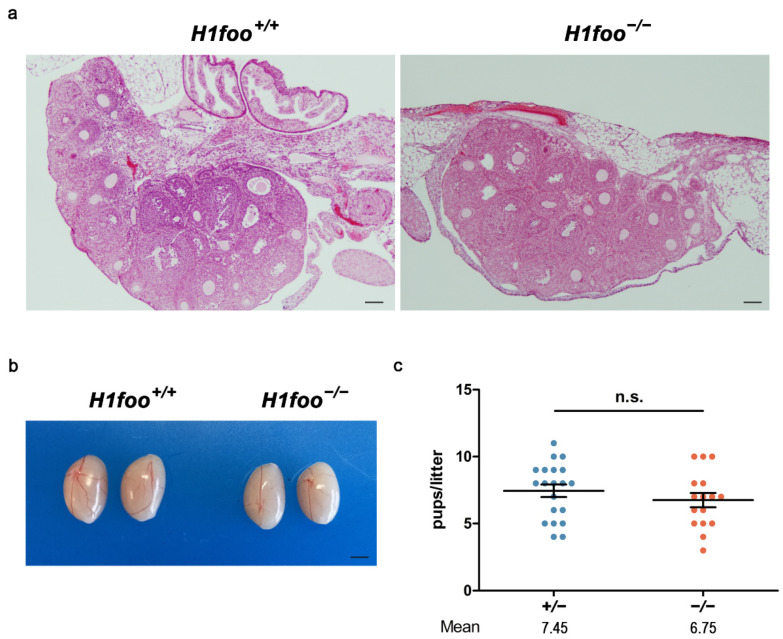
(**a**) Histopathological analysis of hematoxylin-eosin-stained ovaries sections does not show defects in H1FOO-deficient adult mice. Bars in panel, 20 µm. (**b**) Genetic deletion of *H1foo* entails no differences in testis size when compared to their WT counterparts (mice of 3 months of age). Bar in panel, 30 µm. (**c**) Fertility assessment of heterozygous (+/−, blue) and homozygous mutant mice (−/−, red) showing the number of pups per litter. Mean value displayed under the plot. Two-tailed Welch’s *t*-test analysis: n.s., no significant differences.

**Figure 3 cells-11-03706-f003:**
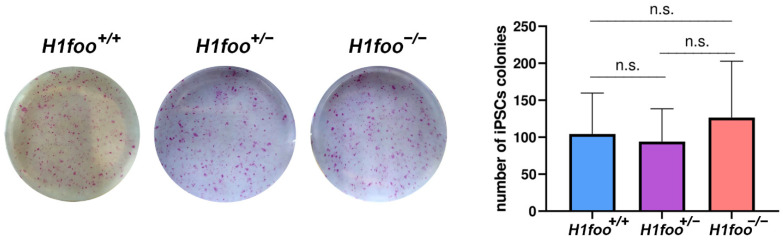
MEFs from the indicated genotypes are infected with the 3 reprogramming factors and the number of alkaline phosphatase positive colonies are counted (plot in the right), showing that the absence of *H1foo* does not significantly affect iPSCs generation. *n* = 12. Welch´s *t*-test analysis: n.s., no significant differences.

## Data Availability

The raw data supporting the conclusions of this article will be made available by the authors, without undue reservation.

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
