# Peer review of "The Oocyte-Specific Linker Histone H1FOO Is Not Essential for Mouse Oogenesis and Fertility"

_cells, 2022, doi:10.3390/cells11223706_

Round 1

Reviewer 1 Report

The authors have generated and examined the mouse model lacking H1FOO. They assessed the fertility, gametogenesis, and progression of prophase I of the H1foo-KO mice. However, regardless of previously expected roles of H1FOO in oocyte development and early embryo, the authors demonstrated that the H1foo-KO male and female mice exhibited no distinct reproductive phenotype and concluded that H1FOO is dispensable during gametogenesis and early embryo development. Although the phenotype of H1foo-KO mice was disappointing, this study provides important insights into the dispensability of oocyte-specific linker histone H1FOO in vivo, especially in gametogenesis, raising a striking impact on the long-awaited understanding of oocyte-specific linker histones in the reproductive epigenetics field. This manuscript is well-written and organized, and potentially interesting, and thus, the manuscript should be open for the field and can be considered for publication if several concerns listed below should be properly addressed.

Comments:

Several previous publications indicated the potential roles of H1FOO in early embryonic development. If the authors could provide data on early embryonic development, such as survival rate of H1foo-KO embryos, this manuscript would be more comprehensive and informative. 

Table S3, Fertility test

To assess the fertility of H1foo-KO mice, the authors examined mating using both male and female H1foo-KO mice. However, this assay cannot exclude the possibility that either H1foo-KO male or female mice are subfertile, which may be true. Therefore, the authors should provide more accurate breeding results. The mating table should compare the breeding results of matings of wild-type or heterozygous male mice with either wild-type/heterozygous or H1foo-KO female mice, along with statistical significance and more than three biological replicates. 

Author Response

We attach our response in the Word file.

Reviewer 2 Report

The manuscript by Sánchez-Sáez et al. described functional analyses of linker histone H1FOO in vivo using H1foo knockout mice, which had been yet to be demonstrated in the field. 

Despite its expression pattern in male germ cells and oocytes, they did not find any significant biological defects in chromosomal dynamics during meiotic prophase, germ cell development and fertility in H1foo knockout mice. The authors also demonstrated that the absence of H1foo did not significantly affect iPSCs generation. Although this study provides negative results regardingH1foo knockout mice, their genetic analyses were through and data were rigid enough to support their conclusion. Therefore, this study will revise the idea on the role of H1foo that was proposed in the previous studies using siRNA and morpholino. Thus, this manuscript should be open to the field. I have some minor comments and suggestions for improving the manuscript as described below. 

Line226-227.  

The authors state that adult H1foo-deficient ovaries were normal, with no alteration in the distribution of follicles and no aberrant appearance of the stroma. This is too strong a word without any quantified data, because appearance of the ovary is different at which section is observed. This should be rephrased. Otherwise, they should present quantification of the distribution and number of follicles in H1foo-deficient ovaries.

Fig1

It is preferable if the authors show western blotting of H1FOO in WT and H1foo-deficient oocytes. 

Fig2b.

A scale bar should be shown.

Author Response

We attach our response in the Word file

Round 2

Reviewer 1 Report

I have looked at the revised manuscript and all is fine. Therefore, I am very pleased to accept the manuscript for publication in Cells.